# Transformer-Based Weed Segmentation for Grass Management

**DOI:** 10.3390/s23010065

**Published:** 2022-12-21

**Authors:** Kan Jiang, Usman Afzaal, Joonwhoan Lee

**Affiliations:** 1Artificial Intelligence Lab, Department of Computer Science and Engineering, Jeonbuk National University, Jeonju 54896, Republic of Korea; 2Division of Computer Science and Engineering, Jeonbuk National University, Jeonju 54896, Republic of Korea

**Keywords:** weed detection, Transformer, semantic segmentation, Swin Transformer, SegFormer, Segmenter

## Abstract

Weed control is among the most challenging issues for crop cultivation and turf grass management. In addition to hosting various insects and plant pathogens, weeds compete with crop for nutrients, water and sunlight. This results in problems such as the loss of crop yield, the contamination of food crops and disruption in the field aesthetics and practicality. Therefore, effective and efficient weed detection and mapping methods are indispensable. Deep learning (DL) techniques for the rapid recognition and localization of objects from images or videos have shown promising results in various areas of interest, including the agricultural sector. Attention-based Transformer models are a promising alternative to traditional constitutional neural networks (CNNs) and offer state-of-the-art results for multiple tasks in the natural language processing (NLP) domain. To this end, we exploited these models to address the aforementioned weed detection problem with potential applications in automated robots. Our weed dataset comprised of 1006 images for 10 weed classes, which allowed us to develop deep learning-based semantic segmentation models for the localization of these weed classes. The dataset was further augmented to cater for the need of a large sample set of the Transformer models. A study was conducted to evaluate the results of three types of Transformer architectures, which included Swin Transformer, SegFormer and Segmenter, on the dataset, with SegFormer achieving final Mean Accuracy (mAcc) and Mean Intersection of Union (mIoU) of 75.18% and 65.74%, while also being the least computationally expensive, with just 3.7 M parameters.

## 1. Introduction

Global population growth has resulted in an increase in food demand. To meet the anticipated demand, the agricultural produce needs to increase by approximately 70% [1]. The farm output and its quality, along with crop cultivation, are, however, adversely affected by a number of factors. Among these issues is the growth of weeds, which occurs simultaneously with crop growth. A variety of weed plants exist that spread quickly and thus negatively impact crop yield. These weeds directly compete with crops for resources such as water, nutrients and sunlight, which leaves the crops prone to a number of diseases. Studies show that the vegetable yield decreases by 45% and up to 95% in the case of weed–vegetable confrontation [2]. This extends even beyond crop cultivation; weed growth is also a problem in turfed surfaces such as those of football and golf, residential lawns, parks and sports fields. In order to tackle the issue of weed gardening, appropriate means must be taken. The focus of this paper is weed control for the latter case of turf grass management, but the same technology might be used for the former case of crop cultivation.

Weed control is a very challenging task. Various strategies can be employed by farmers for weed reduction in targeted areas. These methodologies can be divided into five main categories: (a) preventative—prevent weed growth preemptively; (b) mechanical—mowing, hand weeding and mulching; (c) cultural—maintaining field hygiene; (d) biological—utilizing weeds’ natural adversaries, such as insects, grazing animals, etc.; (e) chemical—spraying herbicides [3]. All of these approaches entail a few drawbacks. These can be either in terms of the required costs/time and crop or environment contamination. As a result, the optimal solution for economic and environmental interests is the development of a vision-based system for the automatic removal of weed plants.

Over the past few years, deep learning (DL) has made huge advancements in multiple domains, including those of vision, audio and text. Object detection and segmentation models designed with the deep learning approach have exhibited high precision in the identification of target objects. As mentioned, weed detection is a very daunting problem, owing to the semantic similarities between weeds and their surrounding background. Common challenges faced during weed detection are similarities in color and texture, occlusion and visual similarities between different weeds. However, advancement in vision-based intelligent machines have made it possible to design an accurate system for the detection of weeds under such complex background conditions.

The target of this work is to propose a deep learning-based model that is able to classify and localize the weed area precisely on grassy fields, i.e., perform weed segmentation, using Visual Transformer [4]. In contrast to weed detection in a crop-filled setting, this work focuses more on detecting weeds in a grassy environment. Owing to the success of the Transformer model in NLP, recent studies have focused on importing it into the visual domain, where it has shown great potential. In the context of weed detection, Youjie et al. [5] have already established the effectiveness of the attention mechanism for the precise segmentation of weeds of varying shapes, and Visual Transformer is an extension of the non-local attention technique.

In this work, we explore the applications of the Transformer model in the context of weed detection and localization. The successful implementation of such a system would greatly reduce the required time and effort for weed identification and removal. This DL-based system should also be robust to a variety of real-life visual challenges, such as deformation, different illumination conditions, occlusion, etc. Such a detection system can be deployed within an autonomous wheeled robot, capable of performing surveillance in the entirety of the grass field and identifying weeds using solely vision. An actuator function similar to weeding by hand might be used to mechanically pull the detected target weeds. Conversely, we can also have sprinklers acting on the exact location of the weed to remove it with a minimum amount of chemicals, making the system eco-friendly and cost/time efficient.

Following our experiments, we report the results of three different types of Transformer architectures, including Swin Transformer [6], SegFormer [7] and Segmenter [8], on our in-house weed dataset. The weed dataset consisted of 1006 images that allowed us to segment 10 types of weeds in grass. The dataset was further augmented to cater for the needs of a large sample set of Transformer models. To increase the trainable sample size in terms of its quality and quantity, we performed a range of different augmentation techniques. The results from these trained models were reported using different metrics. SegFormer achieved the best result on our dataset, with final mAcc and mIoU of 75.18% and 65.74%, respectively. Swin Transformer showed comparable performance to SegFormer, albeit with a much higher number of network parameters. Thus, it may be inferred that the SegFormer-based system would be the most suitable for the automation of weed removal from grassy surfaces.

The keys contributions are two-fold:
Predicting accurate segmentation masks for weeds using Transformer-based architectures for the purpose of automatizing weed control with a focus on turf management.We investigate a range of recent Transformer models using our weed dataset and make detailed comparisons in terms of performance and complexity.

In Section 2, we provide a detailed review of previously designed methods for similar purposes. Section 3 contains information about the Transformer model architectures employed in the study. Section 4 contains details about our dataset, along with the applied augmentations and evaluation metrics. Subsequently, in Section 5, we provide the details of our experiments, the comparison of different models and the extracted conclusions. Finally, in Section 6, we provide a brief summary of the work performed.

## 2. Related Studies

In this section, we provide a brief overview of the Transformer model and its use in computer vision, along with a review of previously proposed vision-based automatic weed detection methods.

### 2.1. Transformer Architecture

Originally introduced in the context of machine translation, Transformer models are now used to solve a wide variety of tasks in multiple domains. In the context of natural language processing, recurrent or convolutional neural network (CNN) models based on encoder and decoder architectures were common before the inception of Transformer models. The Transformer gets rid of the recurrent and convolution layers and proposes a simple model based entirely on the attention mechanism. Transformer models employ the self-attention mechanism, where each word attends to every other word in the same input sequence. As a result, the Transformer model takes significantly less time to train than its counterparts while achieving more parallelization [4].

Building upon the original Transformer architecture, researchers have tried importing the architecture into the domain of computer vision [9,10,11,12]. However, that results in a quadratic cost with reference to the number of pixels, since self-attention in images treats each pixel as a separate token and attends to every other pixel in the image. This, in turn, makes the direct application of self-attention to images practically unfeasible, given the huge number of pixels present in a single image. Various techniques have been designed to mitigate the issue of the quadratic cost of Vision Transformer models. Dosovitskiy et al. [12] applied a pure Transformer block on a sequence of image patches termed Visual Transformer (ViT). In that study, the input image was split into fixed-sized patches, each treated as a single token. The patches were then flattened and underwent trainable linear projection. Positional embedding vectors were added to each input patch, and these patches were then feed-forwarded through a Transformer encoder for classification. Unlike CNNs, in this architecture, the authors did not include any explicit inductive bias about the 2D structure of images, except in the patch extraction and resolution adjustment step. Stand-Alone Self-Attention (SASA) [10] is a fully self-attentive model that replaces all the local convolution operations with self-attention instead of employing self-attention just as an augmentation over convolutions. This substitution operation is performed on the ResNet architecture. Vaswani et al. [13] introduced a new series of self-attention models called HaloNets that are built around the concept of blocked local self-attention, similar to SASA [10]. Swin Transformer [6] is another self-attention-based approach for visual detection. It involves splitting the image into windows of varying sizes between different layers, where self-attention is applied inside these shifted windows. Experiments exploring different locality patterns of the self-attention modules have also been performed [14,15,16].

### 2.2. Deep Learning (DL) Models for Weed Detection and Transformer Models in the Agricultural Sector

Machine learning (ML) has proved to be very effective for the development of automatic weed detection and classification systems for deployment in a wide range of circumstances [17]. Here, we provide a brief overview of previously performed research in this context.

Historically, various image processing techniques were used for the classification of weeds and crops [18,19]. Different shape features are extracted, and the feature vectors are then evaluated using a single-layer perceptron classifier. In contrast to ML techniques that require substantial domain expertise to properly design feature extractors, DL allows the machine to automatically extract the most characteristic features of objects from raw images. DL is more robust, compared with traditional ML models, to different variations in the input images, leading to better classification results.

Espejo-Garcia et al. [20] performed crop (tomato and cotton)/weed (Black nightshade and velvetleaf) identification using a combination of pre-trained convolutional neural networks with traditional machine learning classifiers. Jin et al. [21] identified weeds in a vegetable plantation setting. Contrary to other weed detection systems, their work focused on training a CenterNet model that was first used to detect vegetable and draw bounding boxes around them. Afterwards, the remaining, green-colored objects that fell out of the bounding boxes were classified as weeds. The detection of a large variation in weed species is feasible using this methodology. In their work, the weeds were further extracted from the background using color-index-based segmentation. Vaidhehi et al. [22] developed a model for weed and paddy detection using regional convolutional neural networks (R-CNNs). The results from the R-CNNs were compared with conventional CNN models and other segmentation models. Wang et al. [22] investigated an encoder–decoder based network for the semantic segmentation of crops and weeds. The network was optimized using different input representations. In their experiments, the inclusion of NIR information significantly improved the segmentation accuracy. Youjie et al. [5] combined several techniques with a dilated CNN model to enhance the performance of weed segmentation. They employed hybrid dilated convolution, UFAB (Universal Function Approximate Block), drop-block techniques in the network backbone, bridge attention blocks to link the encoder to the decoder and SPRB (Spatial Pyramid Attention Block) to refine the segmentation result.

Visual Transformer can be treated as an extension of such non-local attention technique. Reedha et al. [23] explored the application of Visual Transformer (ViT) to weed and crop recognition. For this study, the images were collected using a high-resolution camera mounted on an unmanned aerial vehicle (UAV). The UAV was deployed in beet, parsley and spinach fields for dataset collection. Experiments were conducted to compare the effect of varying training and test set sizes. Similarly, Liang et al. [24] used ViT for the classification of soybean and weeds.

Although not directly related to weed detection, there exist studies where ViT models were applied to a diverse set of agricultural problems. In the quest of applying the Transformer model to plant pathology, P. S. Thakur et al. [25] proposed a model named PlantXViT. The proposed model combines the capabilities of traditional convolutional neural networks with Vision Transformer to efficiently identify a large number of plant diseases in several crops. W. Zhu. et al. [26] proposed a method to fuse local and global features of images for feature analysis. They introduced the Transformer encoder as a convolutional operation into the improved model, thereby establishing dependencies between long-distance features and extracting the global features of disease images. The center loss was introduced as a penalty term to optimize the common cross-entropy loss, thus expanding the inter-class differences of crop disease features and narrowing their intra-class gaps. Y. Shen et al. [27] applied Transformer to the field of the semantic segmentation of agricultural aerial images in an attempt to account for the drawback regarding inadequate long-range information utilization associated with fully convolutional networks. A hybrid Transformer (MiT) is employed in the encoder stage to enhance the field anomaly pattern recognition capability, and a squeeze and excitation (SE) module is utilized in the decoder stage to improve the effectiveness of key channels. In order to solve the problems of complex crop disease background and small disease area, [28,29] proposed a lightweight ConvViT model, which combines the convolutional structure and the Transformer structure, and modified the patch embedding method to retain more image edge information for the purpose of facilitating patching information exchange between them. R. Reedha et al. [23] studied ViT for plant classification in unmanned aerial vehicle (UAV) images, demonstrating the potential of ViT for remote sensing image analysis tasks.

The aim of our study is to find the best model for the precise localization and classification of weeds so as to minimize weed removal efforts. To this end, this paper explores the application of recent ViT-based segmentation models, which include Swin Transformer, SegFormer and Segmenter, for the aforementioned purpose.

## 3. Methods

For our experiments, we selected three high-performing Transformer-based segmentation models, Swin Transformer, SegFormer and Segmenter. Public implementations were used for network training. A brief description of each model is provided in the following sections.

### 3.1. Swin Transformer

Swin Transformer is built by replacing the standard multi-head self-attention (MSA) module in a Transformer block with a module based on shifted windows, whereas the other layers are kept the same. As illustrated in Figure 1b, a Swin Transformer block consists of a shifted window-based MSA module, followed by a 2-layer Multilayer Perceptron (MLP) with GELU nonlinearity in between. A LayerNorm (LN) layer is applied before each MSA module and MLP, and a residual connection is also applied after each module. In addition, Swin Transformer also uses the hierarchical feature map constructed by the Patch Merging block to compute the representation of the input. The process of Patch Merging is shown in Figure 2.

As shown in Figure 1, the architecture alternates between Patch Merging and Swin Transformer blocks. Starting off from an input image of size *H* × *W*, the initial Patch Splitting module splits the image into non-overlapping patches, each of which is then treated as a ‘token’ in the input sequence of the split patches. Each patch size is 4 × 4, with a feature dimension of 4 × 4 × 3 = 48. A linear embedding is applied on these raw-pixel valued vectors in order to project it into an arbitrary dimension *C*. Within the whole architecture, the Patch Merging module builds hierarchical feature maps by concatenating the features of each group of 2 × 2 neighboring patches, where the 2 × 2 features within each patch are placed in the channel dimension. This results in a 2× downsampling of resolution. So, the *H*/4 × *W*/4 number of tokens, or patches, is reduced to *H*/8 × *W*/8. The number of tokens is further reduced in the subsequent modules as visualized in Figure 1.

The features coming from the Patch Merging modules are passed through a Swin Transformer block that applies Self-Attention to the partitioned image. The input sequence length is preserved after the application of the attention blocks. Self-attention is implemented in two steps, Window-based Self-Attention (W-MSA) and Shifted Windows Self-Attention (SW-MSA), where these two modules are placed in a sequential manner. In W-MSA, self-attention is applied locally within each window, which leads to a linear increase in complexity with reference to the number of windows or patches. This is an improvement over the previous ViT model, where attention was calculated between each patch/token, which resulted in quadratic complexity with reference to the number of tokens. The SW-MSA approach introduces connections between neighboring non-overlapping windows coming from the previous layer by means of shifting the window configuration slightly.

### 3.2. SegFormer

SegFormer is an efficient semantic segmentation framework based upon the encoder and decoder concepts. The encoder outputs multi-scale features, and a simple All-MLP decoder aggregates this multi-scale information from different layers, combining both local and global attention to compute rich representations in order to perform semantic segmentation.

Figure 3 shows the proposed architecture of SegFormer, which is divided into two sections, the encoder and the decoder. The input image is first divided into 4 × 4 patches, unlike ViT, which uses a patch size of 16 × 16. This results in better performance in dense prediction tasks. The Transformer block in the encoder is composed of three sub-modules: (a) Efficient Self-Attention, (b) Mix-Feedforward Network (FFN) and (c) Overlapping Patch Merging. Efficient Self-Attention is similar to the multi-head self-attention in the original Transformer model; however, it employs a sequence reduction process, as introduced in [7], that results in the reduction in the sequence length using a reduction ratio. This helps to lower the computational cost of the self-attention process. ViT uses fixed resolution Position Encodings (PEs) in order to incorporate positional information, which reduces the performance in the case in which the test and the training resolution differ, since the positional code has to be interpolated for the new resolution. To solve this, SegFormer uses a 3 × 3 Conv in the feed-forward network for data-driven positional encoding. Lastly, the Overlap Patch Merging block is used to reduce the feature map size throughout the architecture. This results in hierarchical feature representation comprising high-resolution coarse features and low-resolution fine-grained features. Hierarchical feature maps of sizes 1/4, 1/8, 1/16 and 1/32 of the original image resolution are obtained as such.

The decoder modules contain a full-MLP layer, which takes the features from the encoder module and aggregates them together. The process is performed in four steps: (a) Multi-level features from the encoder go through an MLP layer to be unified in the channel dimension. (b) The features are then upsampled to 1/4 of their sizes and concatenated together. (c) An MLP layer then concatenates the upsampled features. (d) Lastly, an MLP takes these fused feature maps to predict the final segmentation mask of size *H*/4 × *W*/4 × *N* resolution, where *N* refers to the number of categories.

### 3.3. Segmenter

Segmenter is also a Transformer-based image segmentation model built upon the original Vision Transformer (ViT) that allows modeling global dependencies early on in the architecture. The decoder module of Segmenter is based on the Transformer framework. It adds *K* learnable class embeddings to Mask Transformer, which is input to Transformer as a patch embedding; then, a multiplication operation is performed between the class and the patch embedding, followed by softmax application and 2D feature conversion, with a restoration of the original input image size after upsampling in the end. The final class labels are obtained from these embeddings using a Point-wise Linear decoder or a Mask Transformer decoder. The structure of Segmenter is shown in Figure 4.

The input image, *x* ∈ ***R***^H×W×C^, is first split into a sequence of patches. The raw RGB values are then flattened; then, these vectors are passed through a linear embedding for producing a sequence of patch embeddings. A learnable position embedding is added to the sequence of patches individually for incorporating the location information. These semantic embeddings are then passed through standard Transformer blocks consisting of multi-head self-attention and feed-forward layers to obtain contextualized encoding containing rich semantic information.

This sequence of embeddings is then passed to the decoder, which learns to map these patch-level encodings to patch-level class scores, which are then upsampled using bilinear interpolation to obtain pixel-level scores. This can be performed using a Point-wise Linear decoder or a Mask Transformer decoder. For the Point-wise Linear decoder, a Point-wise Linear layer is applied to the encoder outputs to produce patch-level class logics. This sequence is reshaped into a 2D shape and upsampled to the original image size. Final segmentation maps are obtained by applying softmax to the class dimension. For the Mask Transformer decoder, a set of *K* learnable class embeddings, where *K* refers to the number of classes, are introduced. These are all assigned to a specific semantic class and are used to predict the class map. These class embeddings are processed together with the output embedding of the encoder. The decoder is a Transformer encoder by design that generates *K* masks by computing the scalar product between L2-normalized patch embeddings and the aforementioned class embedding. A set of mask sequences are obtained, which are then reshaped into a 2D mask and upsampled to the original image size. The final segmentation map is obtained after the application of softmax followed by LayerNorm.

## 4. Dataset

As part of the evaluation, we constructed a weed dataset that could be used to assess the model’s performance. The dataset included 10 categories of weeds: clover (Trifolium repens), common ragweed (Ambrosia artemisiifolia), crabgrass (Digitaria), dandelion (Taraxacum), ground ivy (Glechoma hederacea), lambsquarter (Chenopodium album), pigweed (Amaranthus), plantain (Plantago), tall fescue (Festuca arundinacea) and unknown weed. The unknown weed category contained weeds with features different from those of other classes for general weed detection. An example case of every category is visualized in Figure 5, where we can see diverse colors, textures and weed shapes on grassy backgrounds. Note that the density and the colors of grass in the images are different in the cluttered background.

The dataset contained 1006 images in total, as shown in Table 1. All images were taken by lab members using cell phone cameras in Jeonju and Wanju, Jeonbuk Province, in South Korea. As the images were taken in real fields instead of a laboratory, they involved a number of visual challenges, including complex background conditions, differing illumination settings, etc. In addition, the density or the grass growth state varied between different fields. Furthermore, there also existed intra-class variations for each weed class in terms of their color, texture and shape. In Figure 5, we can see complex backgrounds for clover and unknown weed and different illuminations between crab grass and lambs quarter, along with various stages of grass growth in most of the images. In Figure 6, we can find examples of intra-class variations. All these challenges should be dealt with properly to achieve accurate weed segmentation.

Note that such diversity in a training dataset may help to train a model with high robustness, on the condition that its sample size is above a certain threshold. That is part of the reason why the training data should be augmented to enhance the diversity. For our training, we split the dataset into 805 and 201 images for training and testing, respectively.

### 4.1. Data Augmentation

Data augmentation is used to increase the training data to evade overfitting and develop powerful models with limited amounts of initial training samples. However, the results of augmentation should look similar to the images captured in real fields. For augmentation, we used multi-scale training and geometric transforms, including random cropping, random flipping and random rotation, along with photometric distortions, including brightness and contrast changes. Figure 7 and Figure 8 shows some examples of augmented images using geometric transforms and photometric distortions.

In multi-scale training, an original image with size 512 × 512 is randomly changed to a scale of 512–2048 during training. Multi-scale training increases the robustness of the model by training it on images of different sizes.

### 4.2. Evaluation Metrics

We evaluated the semantic segmentation results in terms of two metrics, the pixel accuracy and IoU (Intersection of Union). It is important that the metrics reflect the purpose of weed segmentation. Since the segmentation results can be utilized to control a robot manipulator or to drive a weedicide spray nozzle, the exact localization of the weed area is important in order not to damage any healthy grass.

#### 4.2.1. Pixel Accuracy (PA) and Mean PA (mPA)

The pixels belonging to a class are specified by the target mask, which can be compared with results from test data. The pixel accuracy in a class can be calculated as the ratio of the number of correctly classified pixels to the total number of pixels as
(1)$$\mathrm{PA}=\sum_{i}n_{ii}/\sum_{i}t_{i}$$

The class-wise PA can be averaged over all classes of weed objects to calculate the Mean Average Precision (mAP). Because the exact mask of ground truth for weeds is impossible to specify due to their complicate boundary, the PA can be treated as an approximate to measure the weed area.

#### 4.2.2. Intersection over Union (IoU) and Mean IoU (mIoU)

The IoU is the area of overlap between the predicted segmentation mask and the ground truth divided by the area of union between the predicted segmentation mask and the ground truth. In segmentation, the area is calculated with the number of pixels in a segment. In addition, the object-wise IoUs can be averaged over all objects included in an image to produce the mIoU. From the point of view of its implementation, the IoU for weed objects is important to properly remove the weed using a robot or weedicide to exactly localize the end effector.

In this study, we focused on the weeds that needed to be removed, but the area of background grass is usually much wider than the sparse weed areas, resulting in the mIoU being larger than the IoU of each weed object.

## 5. Results

### 5.1. Implementation Details

For every experiment, we used pre-trained ImageNet weights. For the comparisons, each model implemented in the experiments was the smallest version from its respective family, i.e., Swin Transformer-tiny, SegFormer_mit-b0 and Segmenter_vit-tiny. In Segmenter implementation, the best results were produced using two classes for semantic embeddings, namely, background and weed.

We used a single 2080ti GPU for training with the same training parameters. The AdamW [30] optimizer was chosen with an initial learning rate of 6 × 10^−5^ and a weight decay of 0.01. The scheduler took the linear learning rate decay with a linear warm-up of 1500 and 160 k iterations. For augmentation, we adopted the default settings of random horizontal flip MMSegmentation [31], random rescaling in the ratio range of [0.5, 2.0], random rotation in the range of [0, 360] and random photometric distortion.

### 5.2. Prediction Analysis

The overall segmentation result of each Transformer is summarized in Table 2, whereas the expanded results can be found in Table 3. As shown in Table 2, SegFormer reported the best performance, with the smallest number of parameters, in terms of mIoU and pixel accuracy. On the other hand, Swin Transformer also displayed results comparable to SegFormer.

Table 2 shows the results of each model for every class individually. The weeds, including clover, common ragweed, dandelion, and lambs, had high IoU and pixel accuracy. In contrast, the results on crabgrass and dandelion were comparatively low.

Figure 9a shows the prediction for lambsquarter, where Swin Transformer and SegFormer produced almost perfect segmentation masks, except for the shadowy region, while Segmenter made a mistake on the boundary of the leaf. In Figure 9b, both crabgrass and tall fescue are included in one image. The Swin Transformer and SegFormer made precise masks for the weeds, unlike Segmenter, which failed to produce accurate results. In addition, the segmentation results tried to follow the zigzagged boundary of weed, and even the ground truth mask was smoothly approximated. As shown in the figure, Swin Transformer provided the best result. In Figure 9c, the image contains clover and plantain weeds, and small areas of clover were not included in the ground truth. The results of SegFormer showed that it found out the missing clover areas in the ground truth mask, but Segmenter could not. The results showed that the generalization ability of SegFormer was better than that of Segmenter. In Figure 9d, the ground truth mask only contained ground ivy with a small portion of dandelion towards the lower-left image boundary. Swin Transformer successfully found the dandelion bit that others did not. In general, the object around the boundary is hard to locate or identify, because only limited context information is available to make a proper inference. Swin Transformer was the best in terms of this generalization property with limited context information on the image boundary.

In conclusion, SegFormer produced the best results in terms of IoU and pixel Accuracy of weed objects with the smallest number of parameters, but Swin Transformer was comparable to or better than SegFormer in terms of the generalization ability, while having almost 5× the number of parameters of SegFormer.

## 6. Conclusions

The removal of weeds is essential to successful turf grass management and crop cultivation. Towards this goal, we developed deep learning-based Transformer models to autonomously detect and localize 10 classes of weeds. The dataset introduced in this study includes weed images taken under variable environmental conditions. Case studies were performed on the dataset using three Transformer models, Swin Transformer, SegFormer and Segmenter. The Segmenter model achieved final Mean Accuracy (mAcc) and Mean Intersection of Union (mIoU) of 75.18% and 65.74%. The natural succession to this work is the successful incorporation of the trained models in automated robots for deployment.

## Figures and Tables

**Figure 1 sensors-23-00065-f001:**
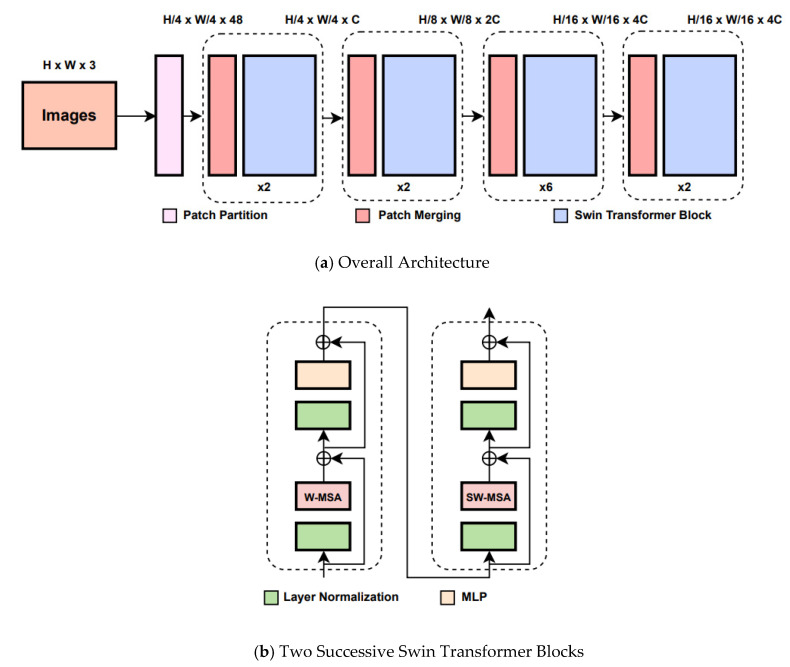
Structure diagram of Swin Transformer.

**Figure 2 sensors-23-00065-f002:**
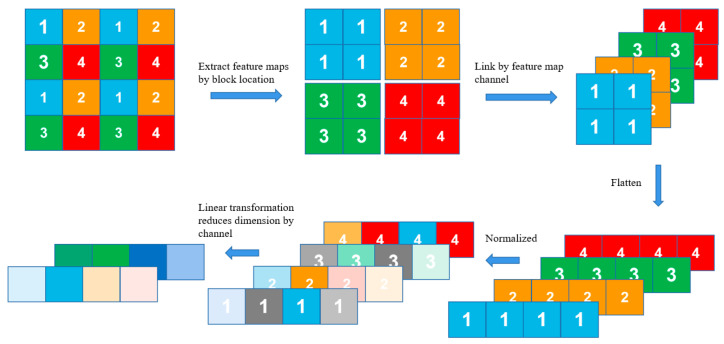
Process of Patch Merging.

**Figure 3 sensors-23-00065-f003:**
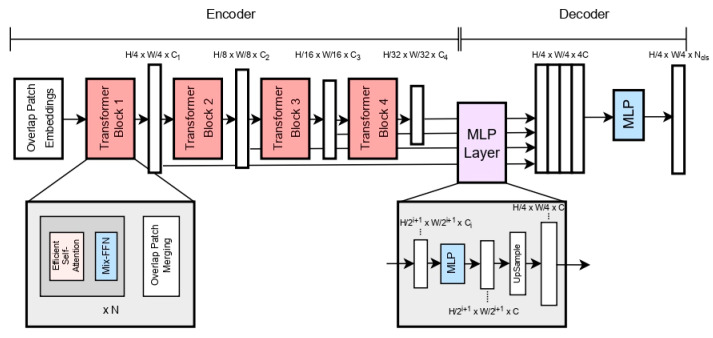
Structure diagram of SegFormer.

**Figure 4 sensors-23-00065-f004:**
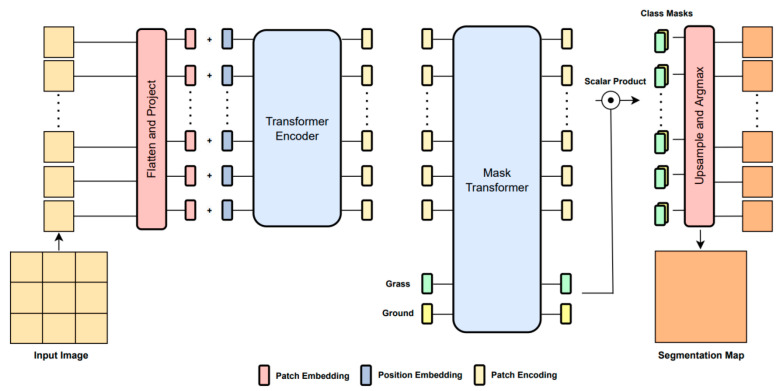
Overall architecture of Segmenter.

**Figure 5 sensors-23-00065-f005:**
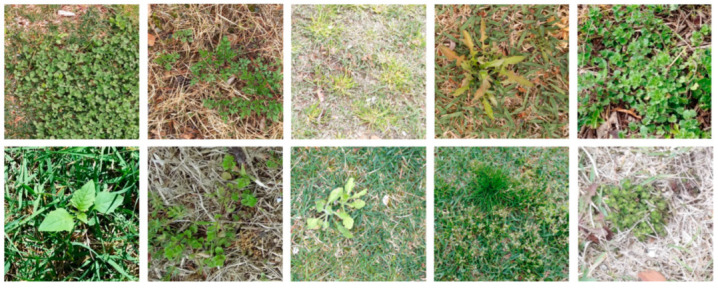
Examples of original images in the weed dataset. From left to right: The first row contains clover, common ragweed, crabgrass, dandelion and ground ivy; the second row contains lambsquarter, pigweed, plantain, tall fescue and unknown weed. (In the sample shown in Figure 5, each picture has only one category of weed, but each picture in the actual collected dataset may have the appearance of multiple types of weeds).

**Figure 6 sensors-23-00065-f006:**
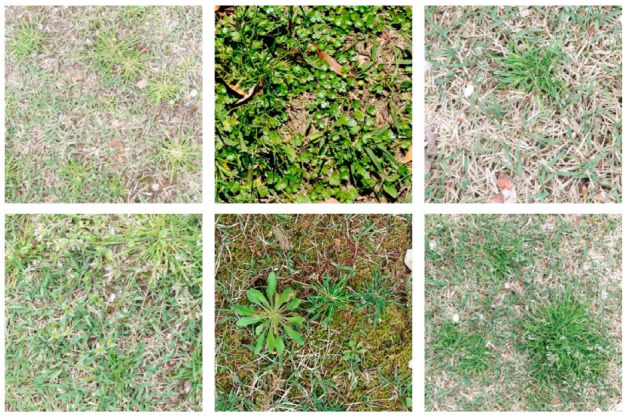
Examples of intra-class (crabgrass) variations.

**Figure 7 sensors-23-00065-f007:**
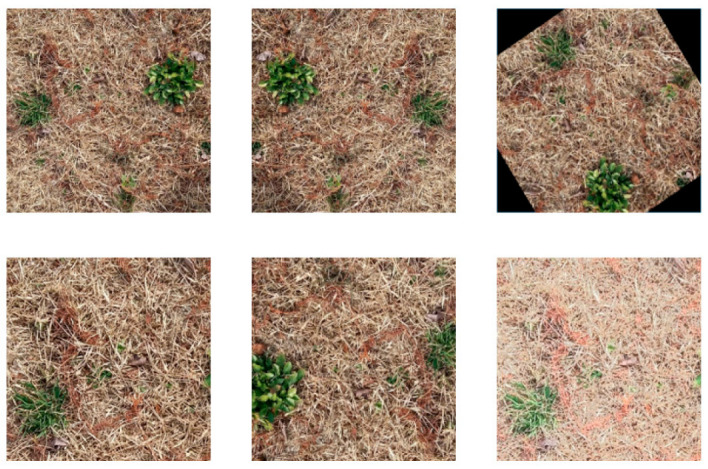
Example images of geometric data augmentation methods (from left to right): original image; horizontal flip; random rotation; random crop; flip + random crop; random crop + photometric distortion.

**Figure 8 sensors-23-00065-f008:**
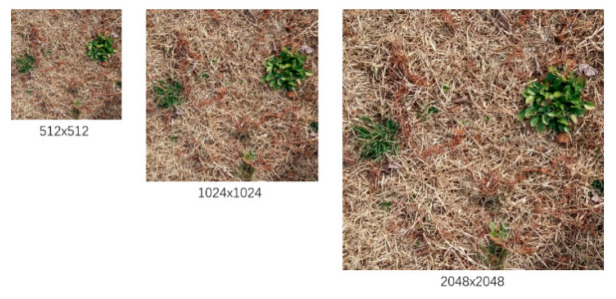
Example results of multi-scale image.

**Figure 9 sensors-23-00065-f009:**
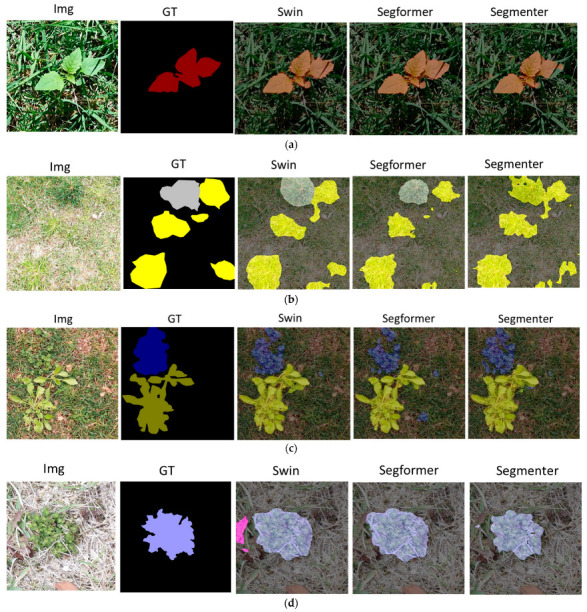
Partial model result visualization: (**a**) results of lambsquarter; (**b**) results of crabgrass and tall fescue; (**c**) results of clover and plantain; (**d**) results of ground ivy and dandelion.

**Table 1 sensors-23-00065-t001:** Image dataset of weeds.

Name of Class	No. of Instances
Clover	717
Common ragweed	105
Crabgrass	219
Dandelion	205
Ground ivy	70
Lambsquarter	55
Pigweed	92
Plantain	451
Tall fescue	175
Unknown weed	93
Total	2184

Table 1: The table shows the total number of weeds contained in the 1006 weed dataset. It can be inferred that the distribution of weed species is not even.

**Table 2 sensors-23-00065-t002:** Experimental results.

	mIoU	mAcc	Param
Swin Transformer	65.41	72.73	29 M
SegFormer	65.74	75.18	3.7 M
Segmenter	59.24	69.31	6 M

**Table 3 sensors-23-00065-t003:** Experimental results.

Class	Swin		SegFormer		Segmenter	
	IoU	Acc	IoU	Acc	IoU	Acc
Background	90.63	97.47	91.74	95.56	90.13	95.13
Clover	78.61	92.28	78.4	91.06	73.36	87.05
Common ragweed	79.15	88.54	82.47	92.53	74.95	83.78
Crabgrass	32.48	34.79	44.26	62.48	40.11	53.76
Dandelion	73.09	82.29	68.91	76.26	63.64	71.79
Ground ivy	86.65	89.05	90.42	97.43	90.75	97.69
Lambsquarter	82.93	88.35	73.86	88.53	76.54	83.76
Pigweed	32.45	36.81	36.1	46.28	22.91	35.29
Plantain	68.71	72.77	63.79	67.57	63.0	68.97
Tall fescue	55.64	72.53	55.41	82.34	51.28	80.12
Unknown weed	39.12	45.14	37.8	42.54	4.9	5.05

## Data Availability

The original contributions presented in the study are included in the article, further inquiries can be directed to the corresponding author.

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
