# Peer review of "Transformer-Based Weed Segmentation for Grass Management"

_sensors, 2022, doi:10.3390/s23010065_

Round 1

Reviewer 1 Report

This study is very interesting. As authors promote one method based on DL to detect the weed. However, I have one question, what is the meaning of this research.

(1) In Figure 9, how do you get these figures, by normal camera or UAV. 

(2) You spend a lot of time to detect weed, however, in (b) and (c), except these detected weed, based on my understanding, the other part mainly also belonged to weed, why don't you detect. 

(3) what's the application of this study. In practice, how do you get the wide-area images with such high resolution? Then, just detect these weeds, then, remove these weeds by manual. So, what's the difference of manual removal directly. In addition, the types of weed are various, how could you guarantee your model's portability.

General comments:

Line 47: There should be These

Line 101: in the next section should be in the next sections

Line 108: suggest not use we in academic manuscripts

Line 150-151: these parts should be introduced in introduction part; Same to Line 158-207.

The English writting really needs improve; the structure also needs to undergone extensive improvement.

Author Response

Dear reviewer,
On the behalf of co-authors, we appreciate editor and reviewers very much for their positive and constructive comments and suggestions on our manuscript entitled “A Transformer-Based Weed Segmentation for Grass Management”. (sensors-2058322).
We have carefully considered reviewer’s comments and have made revision that are marked in red and yellow in the paper. We have tried our best to revise our manuscript according to the comments.
Please find the revised version, which we would like to submit for your kind consideration. Again, we would like to express our great appreciation to you and reviewers for comments on our paper.
Best regards.
Sincerely

Reviewer 2 Report

This research paper addresses the problem of weed detection in several fields, with applicability from vegetable growing to landscape. The problem is of interest and although there have been studies in this direction before, they must be updated and improved.

This is what the authors of this manuscript propose by conducting a study to evaluate the results of three types of Transformer architectures which include Swin Transformer, SegFormer and Segmenter on the dataset. The Segmenter model achieved better results, with a final Mean Accuracy (mAcc) and Mean Intersection of Union (mIoU) of 75.18% and 65.74%.

After reviewing the paper I can say the following:

Title and Abstract clearly state the objective of this experiment.

Introduction is well documented and presents results obtained by other authors, regarding this topic.

I noticed a phrase that doesn’t make much sense, the phrase on row 44-45. Maybe it's a mistake.

Sufficient details regarding methods/process are provided so that other researchers are able to reproduce the experiments described.

I recommend the authors to introduce the Latin name of the weed species they tested, for easier understanding, at least when they first present them, then they can use only the popular name. There are different popular names for the same species of weed but the name in Latin is just one.

The results presented in the conclusion are supported by those by the data, discussed inside the manuscript.

All the references cited relevant and adequate

Author Response

(The authors gave the same response as above.)

Reviewer 3 Report

We should thank the authors for a deep, useful and interesting study that expands the idea of possible modifications of neural network Transformer-Based architectures for identifying and segmenting weeds. The advantage of the article is also a detailed description of the author's dataset used in numerical experiments. As a suggestion, we can recommend the authors to conduct a comparative study on a data set that can be obtained from open sources (for example, Plant Village, etc.).

Author Response

(The authors gave the same response as above.)

Reviewer 4 Report

I suggest using other models since the results are low, you can use transfer learning, Fuzzy or another technique.

Author Response

(The authors gave the same response as above.)

Round 2

Reviewer 1 Report

Authors have addressed my concerns, however, some comments are not well responsed.

(1)The English writting really needs improve; the structure also needs to undergone extensive improvement.

(2)What's the difference of manual removal directly. In addition, the types of weed are various, how could you guarantee your model's portability.

Author Response

Dear reviewer,
On the behalf of co-authors, we appreciate editor and reviewers very much for their positive and constructive comments and suggestions on our manuscript entitled “A Transformer-Based Weed Segmentation for Grass Management”. (sensors-2058322).
Again, we would like to express our great appreciation to you and reviewers for comments on our paper.
Best regards.
Sincerely

Reviewer 4 Report

Overall the article is good

Author Response

(The authors gave the same response as above.)
